# A Proof-of-Concept Protein Microarray-Based Approach for Serotyping of *Salmonella enterica* Strains

**DOI:** 10.3390/pathogens13050355

**Published:** 2024-04-25

**Authors:** Sascha D. Braun, Elke Müller, Katrin Frankenfeld, Dominik Gary, Stefan Monecke, Ralf Ehricht

**Affiliations:** 1Leibniz Institute of Photonic Technology, Member of the Research Alliance “Leibniz Health Technologies’’ and the Leibniz Centre for Photonics in Infection Research (LPI), 07745 Jena, Germany; elke.mueller@leibniz-ipht.de (E.M.); stefan.monecke@leibniz-ipht.de (S.M.); ralf.ehricht@leibniz-ipht.de (R.E.); 2InfectoGnostics Research Campus Jena, Center for Applied Research, 07743 Jena, Germany; 3INTER-ARRAY by Fzmb GmbH, 99947 Bad Langensalza, Germany; kfrankenfeld@fzmb.de (K.F.); dgary@fzmb.de (D.G.); 4Institute of Physical Chemistry, Friedrich Schiller University Jena, 07743 Jena, Germany

**Keywords:** *Salmonella*, protein-based microarray, serotyping, Kauffman–White scheme

## Abstract

*Salmonella enterica*, a bacterium causing foodborne illnesses like salmonellosis, is prevalent in Europe and globally. It is found in food, water, and soil, leading to symptoms like diarrhea and fever. Annually, it results in about 95 million cases worldwide, with increasing antibiotic resistance posing a public health challenge. Therefore, it is necessary to detect and serotype *Salmonella* for several reasons. The identification of the serovars of *Salmonella enterica* isolates is crucial to detect and trace outbreaks and to implement effective control measures. Our work presents a protein-based microarray for the rapid and accurate determination of *Salmonella* serovars. The microarray carries a set of antibodies that can detect different *Salmonella* O- and H-antigens, allowing for the identification of multiple serovars, including Typhimurium and Enteritidis, in a single miniaturized assay. The system is fast, economical, accurate, and requires only small sample volumes. Also, it is not required to maintain an extensive collection of sera for the serotyping of *Salmonella enterica* serovars and can be easily expanded and adapted to new serovars and sera. The scientific state of the art in *Salmonella* serotyping involves the comparison of traditional, molecular, and in silico methods, with a focus on economy, multiplexing, accuracy, rapidity, and adaptability to new serovars and sera. The development of protein-based microarrays, such as the one presented in our work, contributes to the ongoing advancements in this field.

## 1. Introduction

*Salmonella* is a genus of bacteria that can cause foodborne illness. The bacteria can be found in various foods and contaminated water, soil, and animal feces. Symptoms of salmonellosis include diarrhea, fever, abdominal cramps, and vomiting, which can lead to severe cases of dehydration, sepsis, and even death. It is a serious public health concern, with an estimated 95 million cases of gastroenteritis per year caused by *Salmonella* worldwide, and 200,000 cases in Europe [1]. Despite the fact that 90% of salmonellosis cases actually do not require antibiotic therapy, the emergence of antibiotic-resistant *Salmonella* strains complicates the control and management of those infections that do need treatment [2]. There are also examples of resistance genes carried by *Salmonella* strains on mobile genetic elements such as plasmids being transferred to other Gram-negative bacteria in the intestine. Research has shown that antibiotic resistance plasmids can be disseminated between different *Enterobacteriaceae* in the gut, promoting the spread of antibiotic resistance genes [3].

The genus comprises two species, *S. bongori* and *S. enterica*, which are divided into six subspecies and over 2600 serovars based on their somatic (O) and flagellar (H) antigens [4,5]. The different serovars of *Salmonella* are often associated with specific hosts or geographic regions [6,7,8]. Serotyping of *Salmonella* spp. is commonly performed through agglutination testing using the Kauffmann–White scheme, which requires nearly 250 different antisera to identify all serotypes. However, this method only allows the detection of a single antiserum–antigen reaction at a time, it necessitates experienced technologists to perform, and it consumes relatively high volumes of reagents and samples. Due to the need to switch between two different H-phases during the serotyping process (e.g., Sven-Gard agar), it takes a minimum of three days to perform, with a requirement for a minimum of three antiserum–antigen reactions to determine a *S. enterica* serovar [9]. As a result, usually only a few common serovars are tested. Alternative DNA-based approaches, such as PCR, microarray-based assays, or sequencing, have been developed to identify specific serovars [10,11,12,13]. However, PCR methods can only detect a limited number of serovars at a time, and many different assays to analyze molecular markers are yet to be developed or verified for identifying various serovars. In addition, PCR-based mono- and multiplex assays are complex, expensive, and currently still reserved exclusively for specialized laboratories. NGS (next-generation sequencing) has several disadvantages compared to protein microarray-based serotyping of *Salmonella*. It includes the need for bioinformatics expertise, has a much higher cost, longer turnaround time, and it might fail to identify some serovars [14,15]. A research investigation contrasting conventional serotyping with NGS and microarray-based techniques determined that NGS demands specialized skills in bioinformatics and entails a more extended processing duration. In contrast, the microarray approach is resilient and simple to operate, yet it is constrained by a smaller database [16]. Another study highlighted that NGS can lead to major budgetary savings and expeditious result times compared to traditional serotyping, but it requires significant labor and resources [17]. Therefore, while NGS provides detailed genetic information, significant budgetary savings, and reduced labor requirements, it may actually not be as suitable for global routine use as the protein microarray-based method due to the need for specialized expertise and higher costs.

In our research, a proof-of-concept study for a new antiserum microarray-based assay was conducted to demonstrate the technique’s potential for rapid, specific, and sensitive detection of multiple serovars of *S. enterica*. This innovative approach utilizes a panel of carefully selected antibodies targeting unique antigens presented by different serovars, enabling the simultaneous analysis of a wide variety of pathogens within a single experiment. To validate the efficacy of our assay, we include a diverse range of 32 different *S. enterica* subsp. *enterica* serovars and one *S. bongori* in our study. This highly economic platform allows parallel, fast, and economic analysis of multiple antigens investigated for *Salmonella* serotyping.

## 2. Results and Discussion

The results of the experiments are presented in Table 1. Here, the expected results (derived from serology by agglutination) were compared with actual data obtained from protein-based microarrays (all raw data are available in Appendix A). The experiments revealed a concordance rate of 86.00% between the classical serotyping and the microarray results.

For detection, the HRP-labeled PA1-73021 antibody from Invitrogen was consistently used. The O- and H-phases of *S. enterica* serovars Abony (O:4), Agama (O:4), Bredeney (O:4), Budapest (O:4), Choleraesuis (O:4), Gloucester (O:4), Heidelberg (O:4), Saintpaul (O:4), Stanleyville (O:4), and Typhimurium (O:4; microarray is shown in Figure 1A), including the two polyclonal sera, were detected with 100% accuracy (Table 1). Positive signals for both polyclonal sera were also detected for the serovars Brandenburg, Gallinarum, Panama, Stanleyville, and California, which also belong to the O:4 (B) serogroup. No positive signals were obtained for other serovars such as Enteritidis (O:9), Blegdam (O:9), Gallinarum (O:9), Inverness (O:38), Mississippi (O:13), Potsdam (O:7), Nitra (O:2), and *S. bongori* (O:66) when tested with the polyclonal serum TR1101 (O:A-O:67 + Vi). These results showed that this polyclonal serum is not suitable for detecting different serovars as is used in classical serotyping by agglutination. It also shows that this type of assay with all its advantages is possible in principle but must always be optimized for a defined set and combination of reference strains and reference antisera in the course of e.g., a CE-IVD approval. We assume that the concentration of sera other than O:4 is not high enough within one spot to achieve positive signals. Optimization could be achieved by adding additional antisera, utilizing the multiplex capacity of the microarray platform. For example, both O:9 of serovar Enteritidis and O:7 of serovar Kambole should be detected by antisera TR1101. We assume that the concentration of corresponding antisera against O:9 and O:7 in the polyclonal mix TR1101 is too low to obtain positive results with the microarray-based assay. This hypothesis is supported by the results with serum TR1111, where the concentration of some antibodies seems to be higher, resulting in more sensitive detection, and positive signals were seen for the serovars Enteritidis (O:9) and Kambole (O:7), albeit only at the highest concentrations of both the serum TR1111 and the serovar (1:10 dilution of the initial liquid culture).

Antisera detecting O:4, O:5, and O:9, on the other hand, produced positive signals with the corresponding serovars in most cases (Table 1). However, two exceptions were observed: O:9 was negative for serovar Moscow, and O:5 was negative for Paratyphi B. For *Salmonella* Typhimurium, both H-antigens, H1 and H2, could be detected simultaneously, even though normally only one H-phase is formed in the culture [18,19]. For *Salmonella* Heidelberg, both antigens of the H2 phases, H:1,2, could be detected, while the H1:r phase was not tested in the preliminary experiments presented here. Unfortunately, H1:m could not be detected in any of the tested serovars, which could be due to the concentration of the serum being too low or the H1:m phase not being expressed. The detection of the H1:g phase in all corresponding serovars with an accuracy of 100% suggests the latter reason (Table 2). The results belonging to antisera H2:1 and H2:2 were more diverse than expected. We observed good sensitivity and specificity for H2:1, at around 92%, while another scenario was observed for H2:2 with a sensitivity of only 57% (Table 2). According to the Kauffmann–White scheme, all false negatives detected for H2:2 in Montevideo ([1,2,7]), Kambole (1,[2],7), and Stanleyville ([1,2]) were noted as H-factors that may be present or absent without relation to phage conversion [4,20]. These results are in concordance with the sero-genotyping (Appendix A). Negative control experiments with buffer and an *E. coli* strain yielded no signals, indicating the specificity of the capture and detection antibodies.

A comprehensive analysis of the microarray-based assay, considering all spotted antisera collectively, revealed a high degree of accuracy and specificity (Table 2). The overall accuracy of the assay was found to be 86.00%, indicating the ability to correctly classify both positive and negative signals across the array. The overall metrics, as presented in the data, provide a comprehensive snapshot of the collective performance across all evaluated antisera tests. The overall high specificity, recorded at 99.05%, alongside a relatively robust accuracy of 86.00%, underscores the general effectiveness of these tests in correctly identifying individuals who do not possess the target antigens, thereby minimizing the incidence of false positives. This is a crucial attribute, particularly in contexts where the consequences of a false positive result can lead to unnecessary interventions or anxiety. However, a closer examination of the overall sensitivity, which stands at 71.26%, coupled with a Positive Predictive Value (PPV) of 88.67%, suggests that there are significant opportunities for enhancement. The sensitivity metric indicates the tests’ capability to correctly identify true positive cases, i.e., those individuals who actually have the disease or condition. A sensitivity rate of approximately 71.00% implies that nearly 29.00% of true positive cases might be overlooked, which could potentially result in untreated conditions or delayed interventions, impacting patient outcomes adversely. The PPV, while relatively high, further highlights a disparity in the test’s performance. A PPV of 88.67% suggests that when a test yields a positive result, there is an approximately 11% chance that it might not reflect the true disease state. This is particularly significant in low-prevalence settings where the number of false positives could outnumber true positives, thus diminishing the clinical utility of the test in the future. These findings illustrate the assay’s potential for accurate detection, particularly in ruling out negative cases. However, the moderate sensitivity indicates a need for further refinement to enhance its ability to capture all positive cases. Another picture emerged when the polyclonal antisera O:A-67 + Vi and O:A-E + Vi were not included in the calculation of sensitivity and specificity. The overall accuracy in this updated version is markedly higher at 95.36%, compared to 86.00%. Similarly, the NPV increased to 95.50% from 78.89%. Our findings are in line with other recent protein-based and/or molecular-based approaches in *Salmonella* detection [6,21,22]. For example, Cai et al. (2005) [23] developed a protein-based microarray for detecting and discriminating *S. enterica* subsp. *enterica* serotypes. The microarray described by Cai and colleagues also encounters issues in detecting H2:2 and H1:m, especially for serovars Montevideo and Infantis. Additionally, the authors described cross-reactions observed with eight of the 35 antibodies, particularly with O:5. Nevertheless, this described microarray demonstrates potential as a specific detection system for 20 commonly isolated and clinically important serovars.

In summary, all experiments with the proof-of-concept microarray showed promising results. Based on our findings, we plan further research and optimization of the assay principle for the most important *Salmonella* serovars worldwide. Many issues of apparent “false negatives” could easily be solved by spotting additional antisera that were simply not included into the current panel. If developed with sufficient sensitivity and specificity, such an assay could be of great interest to public health authorities and private food control. Additionally, the use of polyclonal antisera such as TR1302 (O:4), TR1406 (H1:g), TR1410 (H1:i), TR1437 (H2:1), and TR1433 (H2:2) in the form of a lateral flow assay could be considered for easy determination of a few *S. enterica* subsp. *enterica* serovars (e.g., Typhimurium) on solid media. Considering this microarray-based assay, which utilizes numerous antisera for *Salmonella* serotyping, it becomes evident that the principle of the microarray could also be effectively utilized for screening and optimization purposes. This offers serum manufacturers a powerful tool for expanding their panels and simultaneously testing all relevant cross-reactivities.

## 3. Materials and Methods

The *Salmonella* antisera (Table 3) were purchased from SIFIN (Berlin, Germany) and diluted to 1:6, 1:8, 1:10, and 1:20 in inhouse buffer 1 (SpoB-1) or protein spotting buffer D12 (SpoB-2) from Scienion (Scienion, Berlin, Germany) (Appendix A). The antisera were then spotted at least four times at a density of 17 × 17 (Appendix A), with a spot distance of 0.22 mm, onto functionalized plastic microarray strips (Scienion, Berlin, Germany) using a fully automated M2 spotter (M2 Automation, Berlin, Germany). The functionalized microarrays facilitated the completion of the coupling reaction within 10 min after printing. The spotter software, utilizing high-resolution images, automatically monitored the spotting quality of all microarrays. All protein-based microarrays were manufactured by INTER-ARRAY by fzmb GmbH (Bad Langensalza, Germany).

*Salmonella* strains of 32 different *S. enterica* subsp. *enterica* serovars and one *S. bongori* (Table 4) were obtained from our small inhouse culture collection. The strains were characterized by agglutination testing according to the Kauffmann–White scheme and genotyped by a DNA-based microarray using the Salm-SeroGenoTyping AS-1 Kit (Abbott, Jena, Germany) according to the manufacturer’s instructions [13]. The original data of the sero-genotyping microarray are available via Appendix A. A scheme of the assay principle is shown in Figure 1B (shown above). The strains were incubated on Columbia Blood agar (BD, Heidelberg, Germany) at 37 °C for 18–24 h. One loop of cells was inoculated directly from the agar into 200 µL phosphate-buffered saline (1× PBS) and vortexed. The arrays were washed twice with 150 µL buffer (1× PBS; 0,05% Tween20; 0.25% TritonX-100; 1% fetal calf serum) for 3 min at 37 °C and 400 rpm using an Eppendorf Thermomixer (Eppendorf, Wesseling-Berzdorf, Germany), followed by 100 µL blocking solution (10% fetal calf serum diluted in 1× PBS; 0.05% Tween20; 0.25% TritonX-100) for 5 min at 37 °C and 300 rpm. In this time, the cell suspensions in PBS were diluted to 1:10 and 1:100. Then, 100 µL of the diluted cells were added to the microarray strip and incubated at 37 °C and 300 rpm for 30 min. The arrays were then washed with 150 µL buffer for 5 min at 37 °C and 400 rpm. For the detection of the specifically bound cells, 100 µL of an HRP-labeled polyclonal anti-*Salmonella* antibody (PA1-73021, Invitrogen by ThermoFisher Scientific, Bremen, Germany) diluted to 1:200 was added and incubated at 37 °C and 300 rpm for 30 min. After a washing step, 100 µL of a precipitating dye was added and incubated for 10 min at room temperature without shaking. Antiserum–antigen reaction signals were scanned under a bright field and automatically analyzed using the INTER-VISION MC Reader and its associated software (version 1.1.0) from INTER-ARRAY by fzmb GmbH. The reader software also analyzed the images and the grey values for each spot automatically. A signal at a particular spot was considered positive at a grey value of 0.1 or more for the corresponding antiserum–antigen reaction. Biotin markers were spotted 12 times as a process control, and these markers were simultaneously used as a microarray orientation matrix.

In the construction of the antiserum arrays, plastic microarray strip substrates from Scienion, functionalized with proprietary Scienion Type 1 chemistry, were utilized. All antibodies in each dilution were successfully covalently immobilized. Two types of spotting buffers, SpoB-1 and SpoB-2 (as mentioned above), were compared; both performed equally well. However, due to missing information regarding the concentration of the polyclonal SIFIN antiserum, the optimal antiserum concentrations were determined by testing serial dilutions. These dilutions were equivalent to antisera dilutions ranging from 1:6 to 1:20 and were used for each antiserum during spotting.

We performed a comprehensive evaluation of the microarray-based serotyping assay by analyzing the data across multiple samples. Each antiserum was assessed for its ability to correctly identify positive (1) and negative (0) signals in a sample set, classified into true positives (TP), true negatives (TN), false positives (FP), and false negatives (FN). For the analysis, we aggregated the results of all spotted antibodies to treat the microarray as a single analytical test. The key performance metrics calculated were Accuracy, Sensitivity, Specificity, Positive Predictive Value (PPV), and Negative Predictive Value (NPV). Accuracy was determined as the proportion of correct predictions (TP and TN) to the total predictions. Sensitivity (true positive rate) and Specificity (true negative rate) measured the test’s ability to correctly identify positive and negative results, respectively. PPV and NPV represented the likelihood that positive and negative results were true positives and true negatives [24].

## Figures and Tables

**Figure 1 pathogens-13-00355-f001:**
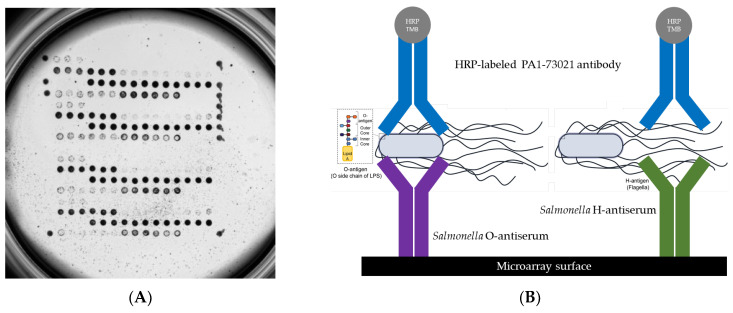
(**A**) Protein-based microarray incubated with *Salmonella enterica* subsp. *enterica* serovar Typhimurium and (**B**) illustration of a sandwich assay on a microarray surface, where *Salmonella* O- or H-antigens are captured between a surface-bound specific antiserum for *Salmonella* and a detection antibody. The detection antibody is conjugated to an enzyme, here depicted as HRP (horseradish peroxidase), which catalyzes a colorimetric reaction with the TMB (tetramethylbenzidine) substrate, producing a measurable signal indicative of the presence of the antigen.

**Table 1 pathogens-13-00355-t001:** Comparative serological analysis of *Salmonella enterica* subsp. *enterica* serovars and *Salmonella bongori* using *Salmonella*-specific polyclonal antisera: assessment of somatic (O) antigen, flagellar antigens H1 and H2, and correlation with fully serologically characterized reference strains for positive (P), negative (N), false positive (FP), false negative (FN), and “true” (meaning: true positive (TP) or true negative (TN)) outcomes. Also, the overall concordance was calculated.

Characteristics of *Salmonella enterica* Serovars, *Salmonella bongori*, and Controls									Antisera		TR1101_A-67_Vi			TR1111_A-E_Vi			TR1302_O:4			TR1303_O:5			TR1307_O:9			TR1406_H1:g			TR1410_H1:i			TS1413_H1:m			TR1437_H2:1			TR1433_H2:2		
				Calculation of Concordance																																				
*Salmonella enterica* Subsp. *enterica serovars* and Controls	Somatic (O) Antigen	Flagellar Antigen H1	Flagellar Antigen H2	Number of Antisera	FN	FN in %	FP	FP in %	True (TP/TN)	Concordance	Expected	Actual	Comparison	Expected	Actual	Comparison	Expected	Actual	Comparison	Expected	Actual	Comparison	Expected	Actual	Comparison	Expected	Actual	Comparison	Expected	Actual	Comparison	Expected	Actual	Comparison	Expected	Actual	Comparison	Expected	Actual	Comparison
Abony	1,4,[5],12,[27]	b	e,n,x	10	0	0%	0	0%	10	100%	**P**	**P**	true	**P**	**P**	true	**P**	**P**	true	**P**	**P**	true	N	N	true	N	N	true	N	N	true	N	N	true	N	N	true	N	N	true
Agama	4,12	i	1,6	10	0	0%	0	0%	10	100%	**P**	**P**	true	**P**	**P**	true	**P**	**P**	true	N	N	true	N	N	true	N	N	true	**P**	**P**	true	N	N	true	**P**	**P**	true	N	N	true
Blegdam	9,12	g,m,q	-	10	3	30%	0	0%	7	70%	**P**	N	**FN**	**P**	N	**FN**	N	N	true	N	N	true	**P**	**P**	true	**P**	**P**	true	N	N	true	**P**	N	**FN**	N	N	true	N	N	true
Brandenburg	4,[5],12	l,v	e,n,z15	10	0	0%	1	10%	9	90%	**P**	**P**	true	**P**	**P**	true	**P**	**P**	true	**P**	**P**	true	N	N	true	N	N	true	N	N	true	N	N	true	N	**P**	**FP**	N	N	true
Bredeney	1,4,12,27	l,v	1,7	10	0	0%	0	0%	10	100%	**P**	**P**	true	**P**	**P**	true	**P**	**P**	true	N	N	true	N	N	true	N	N	true	N	N	true	N	N	true	**P**	**P**	true	N	N	true
Breukelen	6,8	l,z13,[z28]	e,n,z15	10	2	20%	0	0%	8	80%	**P**	N	**FN**	**P**	N	**FN**	N	N	true	N	N	true	N	N	true	N	N	true	N	N	true	N	N	true	N	N	true	N	N	true
Budapest	1,4,12,[27]	g,t	-	10	0	0%	0	0%	10	100%	**P**	**P**	true	**P**	**P**	true	**P**	**P**	true	N	N	true	N	N	true	**P**	**P**	true	N	N	true	N	N	true	N	N	true	N	N	true
California	4,12	g,m,t	[z67]	10	1	10%	1	10%	8	80%	**P**	**P**	true	**P**	**P**	true	**P**	**P**	true	N	N	true	N	N	true	**P**	**P**	true	N	N	true	**P**	N	**FN**	N	**P**	**FP**	N	N	true
Choleraesuis	6,7	c	1,5	10	0	0%	0	0%	10	100%	**P**	**P**	true	**P**	**P**	true	N	N	true	N	N	true	N	N	true	N	N	true	N	N	true	N	N	true	**P**	**P**	true	N	N	true
Corvallis	8,20	z4,z23	[z6]	10	2	20%	0	0%	8	80%	**P**	N	**FN**	**P**	N	**FN**	N	N	true	N	N	true	N	N	true	N	N	true	N	N	true	N	N	true	N	N	true	N	N	true
Cubana	1,13,23	z29	-	10	2	20%	0	0%	8	80%	**P**	N	**FN**	**P**	N	**FN**	N	N	true	N	N	true	N	N	true	N	N	true	N	N	true	N	N	true	N	N	true	N	N	true
Dublin	1,9,12[Vi]	g,p	-	10	2	20%	0	0%	8	80%	**P**	N	**FN**	**P**	N	**FN**	N	N	true	N	N	true	**P**	**P**	true	**P**	**P**	true	N	N	true	N	N	true	N	N	true	N	N	true
Enteritidis	1,9,12	g,m	-	10	2	20%	0	0%	8	80%	**P**	N	**FN**	**P**	**P**	true	N	N	true	N	N	true	**P**	**P**	true	**P**	**P**	true	N	N	true	**P**	N	**FN**	N	N	true	N	N	true
Franken	9,12	z6	z67	10	2	20%	0	0%	8	80%	**P**	N	**FN**	**P**	N	**FN**	N	N	true	N	N	true	**P**	**P**	true	N	N	true	N	N	true	N	N	true	N	N	true	N	N	true
Gallinarum	1,9,12	-	-	10	1	10%	0	0%	9	90%	**P**	N	**FN**	**P**	**P**	true	N	N	true	N	N	true	**P**	**P**	true	N	N	true	N	N	true	N	N	true	N	N	true	N	N	true
Gloucester	1,4,12,27	i	l,w	10	0	0%	0	0%	10	100%	**P**	**P**	true	**P**	**P**	true	**P**	**P**	true	N	N	true	N	N	true	N	N	true	**P**	**P**	true	N	N	true	N	N	true	N	N	true
Goeteborg	9,12	c	1,5	10	2	20%	0	0%	8	80%	**P**	N	**FN**	**P**	N	**FN**	N	N	true	N	N	true	**P**	**P**	true	N	N	true	N	N	true	N	N	true	**P**	**P**	true	N	N	true
Heidelberg	1,4,[5],12	r	1,2	10	0	0%	0	0%	10	100%	**P**	**P**	true	**P**	**P**	true	**P**	**P**	true	**P**	**P**	true	N	N	true	N	N	true	N	N	true	N	N	true	**P**	**P**	true	**P**	**P**	true
Inverness	38	k	1,6	10	2	20%	0	0%	8	80%	**P**	N	**FN**	**P**	N	**FN**	N	N	true	N	N	true	N	N	true	N	N	true	N	N	true	N	N	true	**P**	**P**	true	N	N	true
Kambole	6,7	d	1,[2],7	10	2	20%	0	0%	8	80%	**P**	N	**FN**	**P**	**P**	true	N	N	true	N	N	true	N	N	true	N	N	true	N	N	true	N	N	true	**P**	**P**	true	**P**	N	**FN**
Mississippi	1,13,23	b	1,5	10	2	20%	0	0%	8	80%	**P**	N	**FN**	**P**	N	**FN**	N	N	true	N	N	true	N	N	true	N	N	true	N	N	true	N	N	true	**P**	**P**	true	N	N	true
Montevideo	6,7,14	g,m,[p],s	[1,2,7]	10	5	50%	0	0%	5	50%	**P**	N	**FN**	**P**	N	**FN**	N	N	true	N	N	true	N	N	true	**P**	**P**	true	N	N	true	**P**	N	**FN**	**P**	N	**FN**	**P**	N	**FN**
Moscow	1,9,12	g,q	-	10	3	30%	0	0%	7	70%	**P**	N	**FN**	**P**	N	**FN**	N	N	true	N	N	true	**P**	N	**FN**	**P**	**P**	true	N	N	true	N	N	true	N	N	true	N	N	true
Nitra	2,12	g,m	-	10	3	30%	0	0%	7	70%	**P**	N	**FN**	**P**	N	**FN**	N	N	true	N	N	true	N	N	true	**P**	**P**	true	N	N	true	**P**	N	**FN**	N	N	true	N	N	true
Panama	1,9,12	l,v	1,5	10	1	10%	0	0%	9	90%	**P**	**P**	true	**P**	N	**FN**	N	N	true	N	N	true	**P**	**P**	true	N	N	true	N	N	true	N	N	true	**P**	**P**	true	N	N	true
Paratyphi B	1,4,[5],12	b	1,2	10	2	20%	0	0%	8	80%	**P**	N	**FN**	**P**	**P**	true	**P**	**P**	true	**P**	N	**FN**	N	N	true	N	N	true	N	N	true	N	N	true	**P**	**P**	true	**P**	**P**	true
Potsdam	6,7,14	l,v	e,n,z15	10	2	20%	0	0%	8	80%	**P**	N	**FN**	**P**	N	**FN**	N	N	true	N	N	true	N	N	true	N	N	true	N	N	true	N	N	true	N	N	true	N	N	true
Saintpaul	1,4,[5],12	e,h	1,2	10	0	0%	0	0%	10	100%	**P**	**P**	true	**P**	**P**	true	**P**	**P**	true	**P**	**P**	true	N	N	true	N	N	true	N	N	true	N	N	true	**P**	**P**	true	**P**	**P**	true
Singapore	6,7	k	e,n,x	10	2	20%	0	0%	8	80%	**P**	N	**FN**	**P**	N	**FN**	N	N	true	N	N	true	N	N	true	N	N	true	N	N	true	N	N	true	N	N	true	N	N	true
Stanleyville	1,4,[5],12,[27]	z4,z23	[1,2]	10	1	10%	0	0%	9	90%	**P**	**P**	true	**P**	**P**	true	**P**	**P**	true	**P**	**P**	true	N	N	true	N	N	true	N	N	true	N	N	true	**P**	**P**	true	**P**	N	**FN**
Typhimurium	1,4,[5],12	i	1,2	10	0	0%	0	0%	10	100%	**P**	**P**	true	**P**	**P**	true	**P**	**P**	true	**P**	**P**	true	N	N	true	N	N	true	**P**	**P**	true	N	N	true	**P**	**P**	true	**P**	**P**	true
Uno	6,8	z29	[e,n,z15]	10	2	20%	0	0%	8	80%	**P**	N	**FN**	**P**	N	**FN**	N	N	true	N	N	true	N	N	true	N	N	true	N	N	true	N	N	true	N	N	true	N	N	true
*S. bongori*	66	z41	-	10	1	10%	0	0%	9	90%	**P**	N	**FN**	N	N	true	N	N	true	N	N	true	N	N	true	N	N	true	N	N	true	N	N	true	N	N	true	N	N	true
*Escherichia coli*	6	-	1	10	0	0%	0	0%	10	100%	N	N	true	N	N	true	N	N	true	N	N	true	N	N	true	N	N	true	N	N	true	N	N	true	N	N	true	N	N	true
negative control	-	-	-	10	0	0%	0	0%	10	100%	N	N	true	N	N	true	N	N	true	N	N	true	N	N	true	N	N	true	N	N	true	N	N	true	N	N	true	N	N	true
**Overall Concordance**										**86%**																														

**Table 2 pathogens-13-00355-t002:** Calculation of the diagnostical accuracy, sensitivity, specificity, PPV, and NPV for each *Salmonella*-specific antiserum which was spotted on the microarray surface.

Antisera	Accuracy	Sensitivity	Specificity	PPV	NPV
O:A-67 and Vi	42.86%	39.39%	100.00%	100.00%	9.09%
O:A-E and Vi	54.29%	50.00%	100.00%	100.00%	15.79%
O:4	100.00%	100.00%	100.00%	100.00%	100.00%
O:5	97.14%	85.71%	100.00%	100.00%	96.55%
O:9	97.14%	87.50%	100.00%	100.00%	96.43%
H1:g	100.00%	100.00%	100.00%	100.00%	100.00%
H1:i	100.00%	100.00%	100.00%	100.00%	100.00%
H1:m	85.71%	0.00%	100.00%	0.00%	85.71%
H2:1	91.43%	92.86%	90.48%	86.67%	95.00%
H2:2	91.43%	57.14%	100.00%	100.00%	90.32%
**Overall**	**86.00%**	**71.26%**	**99.05%**	**88.67%**	**78.89%**

**Table 3 pathogens-13-00355-t003:** Polyclonal antisera used for the production of the protein-based microarray to detect important *Salmonella enterica* serovars.

	Order Number	Antiserum	Spotted Dilutions
1	TR1101	Anti-*Salmonella* A-67 + Vi	1:6, 1:8, 1:10, 1:20
2	TR1111	Anti-*Salmonella* I (A-E + Vi)	1:6, 1:8, 1:10, 1:20
3	TR1307	Anti-*Salmonella* O:9	1:6, 1:8, 1:10, 1:20
4	TR1302	Anti-*Salmonella* O:4	1:6, 1:8, 1:10, 1:20
5	TR1303	Anti-*Salmonella* O:5	1:6, 1:8, 1:10, 1:20
6	TR1437	Anti-*Salmonella* H2:1	1:6, 1:8, 1:10, 1:20
7	TR1433	Anti-*Salmonella* H2:2	1:6, 1:8, 1:10, 1:20
8	TR1413	Anti-*Salmonella* H1:m	1:6, 1:8, 1:10, 1:20
9	TR1410	Anti-*Salmonella* H1:i	1:6, 1:8, 1:10, 1:20
10	TR1406	Anti-*Salmonella* H1:g	1:6, 1:8, 1:10, 1:20

**Table 4 pathogens-13-00355-t004:** All tested *Salmonella enterica* subsp. *enterica* serovars and *Salmonella bongori* fully characterized by classical serotyping (agglutination) and sero-genotyping (commercial DNA-based microarray from Alere Inc.). *Escherichia coli* and PBS buffer were included as the negative control.

Number	Name	Subspecies	Somatic (O) Antigen	Flagellar Antigen H1	Flagellar Antigen H2
1	Abony	I	1,4,[5],12,[27]	b	e,n,x
2	Agama	I	4,12	i	1,6
3	Blegdam	I	9,12	g,m,q	-
4	Brandenburg	I	4,[5],12	l,v	e,n,z15
5	Bredeney	I	1,4,12,27	l,v	1,7
6	Breukelen	I	6,8	l,z13,[z28]	e,n,z15
7	Budapest	I	1,4,12,[27]	g,t	-
8	California	I	4,12	g,m,t	[z67]
9	Choleraesuis	I	6,7	c	1,5
10	Corvallis	I	8,20	z4,z23	[z6]
11	Cubana	I	1,13,23	z29	-
12	Dublin	I	1,9,12[Vi]	g,p	-
13	Enteritidis	I	1,9,12	g,m	-
14	Franken	I	9,12	z6	z67
15	Gallinarum	I	1,9,12	-	-
16	Gloucester	I	1,4,12,27	i	l,w
17	Goeteborg	I	9,12	c	1,5
18	Heidelberg	I	1,4,[5],12	r	1,2
19	Inverness	I	38	k	1,6
20	Kambole	I	6,7	d	1,[2],7
21	Mississippi	I	1,13,23	b	1,5
22	Montevideo	I	6,7,14	g,m,[p],s	[1,2,7]
23	Moscow	I	1,9,12	g,q	-
24	Nitra	I	2,12	g,m	-
25	Panama	I	1,9,12	l,v	1,5
26	Paratyphi B	I	1,4,[5],12	b	1,2
27	Potsdam	I	6,7,14	l,v	e,n,z15
28	Saintpaul	I	1,4,[5],12	e,h	1,2
29	Singapore	I	6,7	k	e,n,x
30	Stanleyville	I	1,4,[5],12,[27]	z4,z23	[1,2]
31	Typhimurium	I	1,4,[5],12	i	1,2
32	Uno	I	6,8	z29	[e,n,z15]
33	*Salmonella bongori*	V	66	z41	-
34	*Escherichia coli*	-	6	-	1
35	PBS buffer	-	-	-	-

## Data Availability

All raw data are available via Appendix A as an Excel file and all sero-genotyping reports are available via Appendix A as a PDF file.

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
