# Peer review of "A Proof-of-Concept Protein Microarray-Based Approach for Serotyping of Salmonella enterica Strains"

_pathogens, 2024, doi:10.3390/pathogens13050355_

Round 1

Reviewer 1 Report

Comments and Suggestions for Authors

This study by Braun et al, describes the development of a fast and economic Protein-Microarray based approach for rapid and accurate determination of Salmonella serotypes. The manuscript is pertinent since Salmonellosis is one of the major concerns worldwide. The proper identification of the serotypes involved in outbreaks is highly critical for trace back purpose. Additionally increasing antibiotic resistance in various serotypes of Salmonella posing a public health challenge.

The manuscript is well-written, but the data is too preliminary or limited with only six serovars included in this study. A larger panel of  serovars is recommended. Since Salmonella enterica sub spp enterica alone comprised of over 2500-2600 serovars and about 200-250 have a been involved in outbreaks worldwide.

Below are a few suggested points that will be helpful to improve the quality of the current manuscript before it is accepted for publication..

Line 43: The genus comprises two species, S. bongoris and S. enteritidis should read as genus Salmonella consists of two species: Salmonella enterica and Salmonella bongori.

Line 56 to 74: The authors has identified various drawbacks of other contemporary superior methods but did not explain how much assay would cost and how fast the turn-around time would be? Although the NGS and bioinformatics analysis are pushing button now and do not need expertise and can do subtyping of problematic serovars like enteritidis, Heidelberg and Typhimurium which is not possible by this developed method or even the White Kauffmann scheme.

Line 75-78: Table 1 is totally flawed as these are not Salmonella enteritidis serovars instead Salmonella enteritidis is one of the serovar of Salmonella enteric sub sp. enterica. It occurred to me the authors is not familiar with the true Salmonella classification and Nomenclature which appears to be a big problem.

 Line 83: The concordance rate of 88.75% between the classical serotyping and the microarray results is not acceptable and only 5 to six serovars have been tested.  Sensitivity value is also low and not acceptable. The author should include over 100 serovars in this study to validate their method. The results seem too preliminary and method implementation is not feasible with such a small study.

Line 99: The authors have cited Cai et al. (2005) but this is not a true comparison with this study as Cai et al included a larger data set.

The supp file is not accessible.

I will stop my review here and recommend making corrections and include more strain and revalidate their method. Inclusion/exclusion control missing in the experiments.

Comments on the Quality of English Language

Need some improvement

Author Response

Dear Reviewer,

Thank you very much for taking the time to review our manuscript. Please find the detailed responses attached. The corresponding revisions and corrections are highlighted and tracked in the resubmitted files.

Sincerely,

Sascha Braun

Reviewer 2 Report

Comments and Suggestions for Authors

The manuscript by Braun et al. entitled as "A fast and economic protein-microarray based approach for serotyping of Salmonella enterica strains" is devoted to the testing of the protein-microarray based approach for Salmonella serotyping. The authors very briefly described the method and results of the protein microarray method for serotyping 6 of more than 2500 Salmonella serovars. Although the method has many disadvantages, the authors believe that it is quite cost effective compared to the classical agglutination and multiplex PCR serotyping methods. Since English is not the native language of the authors, I strongly recommend a thorough English editing.

Due to the limited length of the manuscript (the results combined with the discussion take up only one and a half pages out of eight), I would recommend that it be labeled as a short communication.

In addition, I would recommend that the authors choose one of the terms, either serotype (as in the abstract and part of the text) or serovar for Salmonella, and use it throughout.

Other minor comments:

L41: Enterobacteriaceae in italics.

L43: It is obvious that the authors are not very familiar with Salmonella nomenclature. The correct species names are S. enterica and S. bongori, and the number of subspecies is six, not five. Finally, the number of serovars/serotypes is over 2700.

The tables are chaotically arranged. They should be on the next page after the reference, not before (the same applies to figures). Numbering should be in order. In the text, after Table 1 (L80), Table 3 (L91) is mentioned, not 2. Similarly, Table S1 (L82) is followed by Table S4 (L112).

Table 1: Salmonella enterica is in the header. Escherichia coli in full on first mention.

L91: Typo: from

Figure1B: Salmonella in italics.

L128: The failure to detect the very important H1:m antigen is extremely poor, especially since it differentiates between major serovars such as Enteritidis and Dublin.

L131: Typo: Supplementary.

L162: I would like more data on the Salmonella tested: Collection numbers, year and place of isolation, origin (clinical or environmental), if they had whole genome sequencing. How were they stored, at what temperature, how were they subsequently cultured, on what media, were they passaged on broth, how many passages? The expression of antigens, especially the H1 and H2 phases, depends on all this.

L166: Typo: principle

L213: Escherichia coli in italics.

Comments on the Quality of English Language

Since English is not the native language of the authors, I strongly recommend a thorough English editing.

Author Response

Dear Reviewer,

Thank you very much for taking the time to review this manuscript. Please find the detailed responses attached. The corresponding revisions and corrections are highlighted and tracked in the resubmitted files.

Sincerely,

Sascha Braun

Round 2

Reviewer 1 Report

Comments and Suggestions for Authors

The quality of the manuscript has been improved compared to earlier version. The data presented as proof of concept study is adequate and good quality.

Author Response

Dear Reviewer,

Thank you for your positive feedback on the revisions of our manuscript with the ID pathogens-2915423. We greatly appreciate your recognition of the improvements we made and are glad to hear that the data presented meets the standards of a proof of concept study. Your insights and detailed feedback have been instrumental in enhancing the quality of our work. We are grateful for the time and effort you have taken to review our manuscript.

Sincerely,

Sascha Braun

Reviewer 2 Report

Comments and Suggestions for Authors

The authors have done considerable work in editing the manuscript.

Regarding the number of serovars, I was able to find a reference that mentions 2,659 serovars (appendix number 48 of the Kaufmann-White-Le Minore scheme): https://www.sciencedirect.com/science/article/pii/S0923250814001065.

Also, in line 30, you should remove "(S.)". This is the obvious thing.

Comments on the Quality of English Language

English editing is sufficient for publication.

Author Response

Dear Reviewer,

I hope this message finds you well. First and foremost, I would like to thank you for the thorough review of my manuscript with the ID pathogens-2915423. Your constructive comments and valuable insights have significantly contributed to improving the quality of our work, for which I am truly grateful.

Comment 1:

Regarding the number of serovars, I was able to find a reference that mentions 2,659 serovars (appendix number 48 of the Kaufmann-White-Le Minore scheme): https://www.sciencedirect.com/science/article/pii/S0923250814001065.

Authors: Thank you for directing us to the specific reference detailing the number of serovars as per the Kaufmann-White-Le Minore scheme. We have duly updated line 43 of our manuscript to include the precise figure of "over 2600 serovars," which is now supported by the citation you provided in appendix number 48. Additionally, we have incorporated the recommended literature into line 44 of our script.

Comment 2:

Also, in line 30, you should remove "(S.)". This is the obvious thing.

Authors: Thank you for pointing out the unnecessary notation in line 30 of our manuscript. We agree that including "(S.)" may be redundant, and we have removed it as per your suggestion. We appreciate your attention to detail and your guidance in refining our manuscript. Thank you once again for your helpful feedback.

Sincerily,

Sascha Braun